# Validation of Four Thyroid Ultrasound Risk Stratification Systems in Patients with Hashimoto’s Thyroiditis; Impact of Changes in the Threshold for Nodule’s Shape Criterion

**DOI:** 10.3390/cancers13194900

**Published:** 2021-09-29

**Authors:** Dorota Słowińska-Klencka, Mariusz Klencki, Martyna Wojtaszek-Nowicka, Kamila Wysocka-Konieczna, Ewa Woźniak-Oseła, Bożena Popowicz

**Affiliations:** 1Department of Morphometry of Endocrine Glands, Medical University of Lodz, Pomorska Street 251, 92-213 Łódź, Poland; kamilawysocka@tyreo.umed.lodz.pl (K.W.-K.); ewawozniak@tyreo.umed.lodz.pl (E.W.-O.); bozena.popowicz@umed.lodz.pl (B.P.); 2Department of Clinical Endocrinology, Medical University of Lodz, Pomorska Street 251, 92-213 Łódź, Poland; martyna.wojtaszek-nowicka@umed.lodz.pl

**Keywords:** thyroid, nodule, cancer, ultrasound, Thyroid Imaging Reporting and Data Systems (TIRADS)

## Abstract

**Simple Summary:**

Thyroid Imaging Reporting and Data Systems (TIRADS) optimize the selection of thyroid nodules for cytological examination. There is a question: is the effectiveness of these systems affected by morphological changes to thyroid parenchyma that are visible in the course of Hashimoto’s thyroiditis (HT)? This question is very important because of the increased risk of malignancy in thyroid nodules in patients with HT. We investigated widely accepted ultrasound malignancy risk features with a special consideration of the suspected nodule’s shape in patients with and without HT. We also validated EU-TIRADS, K-TIRADS, ACR-TIRADS, and ATA guidelines in both groups and evaluated the impact of changes in the threshold for nodule’s shape criterion on the diagnostic value of these TIRADS. The presence of Hashimoto’s thyroiditis did not exert any significant adverse implications for the efficiency of examined TIRADS. The impact of changes in the threshold for nodule’s shape criterion was the highest for EU-TIRADS.

**Abstract:**

The aim of the study was to validate thyroid US malignancy features, especially the nodule’s shape, and selected Thyroid Imaging Reporting and Data Systems (EU-TIRADS; K-TIRADS; ACR-TIRADS, ATA guidelines) in patients with or without Hashimoto’s thyroiditis (HT and non-HT groups). The study included 1188 nodules (HT: 358, non-HT: 830) with known final diagnoses. We found that the strongest indications of nodule’s malignancy were microcalcifications (OR: 22.7) in HT group and irregular margins (OR:13.8) in non-HT group. Solid echostructure and macrocalcifications were ineffective in patients with HT. The highest accuracy of nodule’s shape criterion was noted on transverse section, with the cut-off value of anteroposterior to transverse dimension ratio (AP/T) close to 1.15 in both groups. When round nodules were regarded as suspicious in patients with HT (the cut-off value of AP/T set to ≥1), it led to a three-fold increase in sensitivity of this feature, with a disproportionally lower decrease in specificity and similar accuracy. Such a modification was effective also for cancers other than PTC. The diagnostic effectiveness of analyzed TIRADS in patients with HT and without HT was similar. Changes in the threshold for AP/T ratio influenced the number of nodules classified into the category of the highest risk, especially in the case of EU-TIRADS.

## 1. Introduction

Preoperative diagnostics of thyroid nodules is based on two main pillars—ultrasound imaging (US) and fine needle aspiration biopsy (FNA). The ultrasonographic examination is mainly used for the assessment of ultrasound malignancy risk features (US malignancy features), and subsequent qualification of nodules into particular categories of sonographic risk stratification systems (SRSs). These systems are usually called TIRADS (Thyroid Imaging Reporting and Data Systems) and they enable a more efficient estimation of the risk of malignancy (RoM) in nodules than the evaluation of separate US malignancy features. The most popular SRSs include EU-TIRADS—recommended by European Thyroid Association (ETA), K-TIRADS—recommended by Korean Society of Thyroid Radiology (KSThR), ACR-TIRADS—created by American College of Radiology (ACR), and the system recommended by American Thyroid Association (ATA guidelines) [1,2,3,4]. Our analyses, as well as many reports from other centers, indicate that all these systems not only optimize the selection of nodules for cytological examination but also help to make clinical decisions in patients with an equivocal FNA outcome [5,6,7,8,9]. There is, however, some disagreement between TIRADS systems about the precise definition of particular US malignancy features and their optimal association. One of the areas of notable difference refers to the definition of the nodule’s suspicious shape, usually described as ‘more taller than wide’. Not all SRSs include precise instructions on how to categorize nodules with the anterior-posterior (AP) dimension equal to the transverse (T) dimension or which thyroid plane should be used for the shape evaluation. Remarkably, there are reports that suggest a rationale for adopting a larger than 1.0 threshold for the AP to T ratio [10], and even studies indicating that the optimal threshold should be <1 [11,12]. There is another important question: is the effectiveness of the suspected nodule’s shape and other US malignancy features affected by morphological changes to thyroid parenchyma that are visible in the course of Hashimoto’s thyroiditis (HT).

HT is the most common autoimmune endocrine disease, as well as the most common cause of hypothyroidism. This inflammation is characterized by a progressive loss of thyroid follicular cells and lymphocytic infiltration of the thyroid parenchyma associated with fibrosis [13,14]. It is usually accompanied by a decrease in the gland’s volume and several characteristic changes visible on US imaging. The thyroid may be hypoechoic, with a coarse, heterogeneous parenchymal echotexture, or have the presence of the marginal abnormality, echogenic septations, multiple discrete hypoechoic micronodules or pseudo-nodular structures. These features may be present separately or in different sets and make it difficult to differentiate between thyroid nodules and pseudonodules, and in the former case—between cancers and benign lesions [15,16,17]. The latter problem is of particular importance considering the increased risk of papillary thyroid carcinoma (PTC) in the case of nodules coexisting with HT [18,19].

Thus, we decided to validate US malignancy features, with a special consideration for the nodule’s shape, and selected TIRADS systems in patients with or without coexisting HT (HT and non-HT groups).

## 2. Results

### 2.1. Effectiveness of the Assessment of Suspicious Nodule’s Shape in Differentiation between Benign and Malignant Nodules in HT and non-HT Groups

The usefulness of AP/T ratio assessment in the differentiation between benign nodules and cancers, as measured with area under the receiver operating characteristic curve (AUC), was similar in both groups (transverse plane, Z: −0.1893, *p* = 0.8498; longitudinal plane, Z: 0.2837, *p* = 0.7767) (Table 1). When indexes of diagnostic effectiveness were calculated for the threshold AP > T they were found to be nearly the same in the case of transverse plane. However, in the case of longitudinal plane the AP > T threshold was ineffective in patients with HT.

When AP > T threshold was replaced with AP ≥ T one, a significant increase in sensitivity (SEN) was observed in both groups, and that increase was higher in HT group than in non-HT group. More pronounced changes in HT group were a consequence of the higher incidence of round nodules with AP = T in that group in comparison with non-HT group (Appendix A). On the transverse plane it was the case for both benign nodules and cancers. On the longitudinal plane the differences were smaller, insignificant and they were observed only for cancers. The higher incidence of round cancers on transverse plane was observed not only for PTC (HT: 27.6% vs. non-HT: 16.3%, *p* = 0.0408), but also for other malignant nodules, although insignificantly (HT: 27.3% vs. non-HT: 10.4%, *p* = 0.1408). Consequently, the threshold AP ≥ T on transverse plane was the only effective threshold in HT group for revealing cancers other than PTC, odds ratio (OR): 4.5, CI 95%: 1.3–15.6, *p* = 0.0160.

The highest accuracy (ACC) values for the differentiation between benign and malignant nodules were noted in both groups in the case of transverse plane. Maximal accuracy was reached in HT group with the cut-off value of AP/T ratio set to 1.14, while in non-HT group—to 1.17 (Appendix A). With those thresholds changes in SEN and specificity (SPC) did not exceed 4% in both groups when compared with the threshold AP > T, while risk of malignancy (RoM) of nodules was about 20% higher, and positive likelihood ratio (LR+) and OR increased twofold (Table 1). There were no significant differences between examined groups in indexes of diagnostic effectiveness of the suspicious shape when thresholds optimized for ACC were used.

When the longitudinal plane was used for measurements, the maximal ACC values were slightly lower than in the case of transverse plane and the optimal cut-off value of AP/T ratio was found to be AP ≥ T in both groups (Appendix A). Regardless of the adopted cut-off value, no improvement in ACC values was observed in any of the groups when the assessment of nodule’s shape was performed on both planes (with positive nodules defined as those reaching the threshold on any plane) in comparison with the assessment on a single plane (Appendix A).

There was no significant difference in AUC for nodules <1 cm and larger ones in either group (Z: 1.0524, *p* = 0.2926; non-HT: Z: −0.6656, *p* = 0.5056), but in HT group the assessment of suspicious shape feature was ineffective in nodules <1 cm (Appendix A). In non-HT group significant differences in the frequency of nodules <1 cm with suspicious shape between cancers and benign nodules were observed only for the threshold AP ≥ T.

### 2.2. Effectiveness of the Assessment of Other US Malignancy Features

In non-HT group almost all other US malignancy features, except for pathological vascularization and rim calcifications, were observed significantly more often in cancers than in benign nodules (Table 1 and Table 2). In the case of HT group, the list of insignificant features included also solid echostructure, more solid than cystic echostructure, and macrocalcifications. The logistic regression analysis showed that in HT group the presence of microcalcifications was the strongest indication of nodule’s malignancy (OR: 22.7), and that the presence of suspicious margins or marked hypoechogenicity increased the risk of malignancy at least tenfold. In non-HT group, the strongest index of nodule’s malignancy was the presence of irregular margins (OR: 13.8).

Microcalcifications, irregular margins, marked hypoechogenicity, suspicious shape and hypoechogenicity were independent features in the differentiation between benign and malignant nodules in both groups (Appendix A). Macrocalcifications and solid echostructure were such features only in non-HT group.

Benign nodules of HT group were solid significantly more often than those of non-HT group (91.1% vs. 70.1%, *p* < 0.0001) and more solid than cystic (97.4% vs. 85.7%, *p* < 0.0001), but they showed pathological vascularization less frequently (13.7% vs. 19.1%, *p* < 0.0474). Malignant nodules of HT group contained microcalcifications more often than cancers of non-HT group (29.9% vs. 18.3%, *p* < 0.0273), while macrocalcifications were less frequent (5.8% vs. 19.2%, *p* < 0.0033). Spongiform echostructure was observed only in benign nodules of both groups, but it was less common in HT group than in non-HT group (5/1.9% vs. 46/7.4% respectively, *p* = 0.0018).

### 2.3. Comparison of the Effectiveness of Analyzed SRSs

Table 3 shows the distribution of benign and malignant nodules among particular categories of the examined SRSs for each of previously analyzed thresholds of AP/T ratio with measurements on transverse plane. In the majority of cases, calculated RoM for particular categories of analyzed TIRADSs corresponded to expected RoM or differed by less than 5%. Larger differences (up to 10%) were observed in the case of lower than expected RoM for category 5 of EU-TIRADS with the threshold AP ≥ T in both groups as well as higher than expected RoM for category 4 of K-TIRADS, ACR-TIRADS and ATA guidelines SRS in non-HT group.

All analyzed SRSs showed the highest ACC of distinguishing between benign and malignant nodules when category 5 was used as a cut-off level, irrespectively of the adopted threshold of AP/T ratio. Table 4 shows values of indexes describing the effectiveness of that distinction. System EU-TIRADS, with any of analyzed thresholds of AP/T ratio, showed the highest SEN and negative predictive value (NPV), but the lowest SPC and positive predictive value (PPV) (see Table 3). On the other hand, ACR-TIRADS system was characterized by the lowest SEN and the highest SPC. Generally, we did not find significant differences in AUC between analyzed SRSs at the same AP/T ratio thresholds. The only exception was observed in non-HT group, where AUC for ACR-TIRADS was significantly lower than for other SRSs when the threshold was AP/T ≥ 1.17. No significant differences were found in the effectiveness any of SRSs between HT and non-HT groups when the comparison was made at the same AP/T ratio threshold.

When AP > T threshold was replaced with AP ≥ T one, the numbers of nodules classified into category 5 increased in all SRSs. Accordingly, there was an increase in SEN of that category (when it was used as a threshold for malignancy) and a decrease in its SPC and RoM (Table 3 and Table 4). In the case of EU-TIRADS that effect was stronger in HT group than non-HT group. In the former group the number of nodules in category 5 increased by 32.6%, SEN increased by 11.2%, SPC decreased by 11.2% and RoM decreased by 16.1%. In non-HT group analogous changes were 15.7%, 7.0%, 4.7% and 7.4%, respectively. In other SRSs the resultant changes in SEN, SPC, RoM and the percentage of nodules classified into category 5 were similar in both groups (Appendix A).

When the AP/T ratio threshold was optimized to obtain the maximal ACC (HT group: AP/T ≥ 1.14; non-HT group: AP/T ≥ 1.17) the most distinct effects in comparison to AP > T threshold were observed again in the case of EU-TIRADS. They were especially visible in non-HT group, where RoM increased by 7.2% and SEN decreased by only 0.8%. AUC value in non-HT group for EU-TIRADS with the threshold AP/T ≥ 1.17 was significantly higher than for AP > T threshold. No significant difference in AUC was observed in any other SRSs when AP/T ratio threshold was changed in either group.

## 3. Discussion

Hashimoto’s thyroiditis is a common thyroid disease, especially in areas of high iodine supply. It is usually accompanied by significant changes in the morphology of the gland that impair the identification of thyroid nodules and the assessment of a nodule’s US malignancy features. These difficulties are commonly aggravated by the small size of nodules. Despite these complications, our analysis shows that all four of the most recognized and strongest US malignancy features (i.e., marked hypoechogenicity, irregular margins, microcalcifications, and suspicious shape) are effective in distinguishing benign nodules from cancers in the case of coexisting HT. Other investigators also indicate that the majority of US malignancy features present similar efficiency in patients with or without HT [16,20,21,22,23,24,25,26]. However, there are some differences. In our study, microcalcifications were an almost three times stronger indication of malignancy in a nodule in patients with HT than in those without HT. We believe that this difference is a consequence of increased prevalence of PTC among cancers in patients with HT. In addition, it is PTC that microcalcifications are particularly characteristic of. However, there is no full agreement on the incidence of various types of calcifications in patients with HT. As with our observations, Baser et al. (2015) found that macrocalcifications were observed less often in HT than non-HT patients [22]. Durfee et al. (2014) and Gul et al. (2010) did not find any significant differences in calcification types between HT and non-HT groups [20,25]. In addition, Ohmori et al. (2007) even noted an increased incidence of dense calcifications and decreased incidence of psammoma bodies in thyroid cancer associated with HT compared to cancers without HT [27]. More concordant is the opinion that the assessment of solid structure of nodules is not useful in patients with HT [16,23,24,26], as can also be concluded from our study. That is a consequence of the fact that almost all nodules which accompany HT are solid, even the benign ones. This may result from their smaller sizes; an influence of morphological changes induced by HT cannot be excluded either. Observations regarding suspicious margins are less concordant [16,20,28]. In our study, their assessment was effective regardless of the presence of HT. Such results can be achieved only if the assessment is performed by an experienced ultrasonographer. Experience is necessary to avoid the interpretation of ill-defined pseudonodules as true nodules. Our team is fully aware of that danger due to our previous studies on the relation between FNA outcomes in patients with HT and different ultrasonographic images of thyroid parenchyma, including variants with pseudonodules [29]. All systems analyzed in our study regard irregular margins as suspicious. The authors of K-TIRADS underline that ill-defined margins are visible not only in thyroiditis, but also in infiltrative malignant tumors, but in our opinion the latter almost always present irregularity in their margins. The problem arises because that irregularity is frequently very fine, which gives an impression of ill-defined margins.

The most complicated issue related to US malignancy features is the assessment of the suspicious shape of a nodule. We have shown that thyroid nodules in patients with HT are round (their AP = T) more often than nodules in patients without HT. When the transverse plane is used for measurements the above is equally true for benign nodules and cancers, and in the latter case, not only PTCs but other cancers too. When the longitudinal plane is used, only cancers are more often round. Consequently, in patients with HT the assessment of the suspicious shape of a nodule on the transverse plane is effective with both thresholds: AP > T and AP ≥ T, while on the longitudinal plane only AP ≥ T threshold is effective. Interestingly, when AP ≥ T threshold is used on the transverse plane the assessment of the suspicious shape becomes also effective in diagnosis of cancers other than PTC. Previously, other authors did not find the assessment of suspicious shape to be useful for diagnosing patients with FTC [30], so that issue should be investigated further. However, the key question is an equivocal definition of the suspicious shape of a nodule. It is usually described as ‘more taller than wide’, which in practice in the majority of studies translates into AP > T threshold, but in some of them into the AP ≥ T one [31]. In the guidelines and in many reports, there are no precise indications as how to categorize round nodules or such indications are contradictory. Among SRSs analyzed in our study, EU-TIRADS identifies the suspicious shape of a nodule as non-oval or round [1]. In contrast, K-TIRADS defines the suspicious shape as neither round nor oval [2]. Similarly, in ACR-TIRADS the threshold of AP/T ratio >1, and ATA guidelines use the description ‘more taller than wide shape’ [3,4]. There is also some controversy over the plane of measurements. K-TIRADS specifies that AP > T should be observed on a transverse or longitudinal plane, while ACR-TIRADS and ATA guidelines limit that condition to the transverse plane only. In addition, EU-TIRADS does not specify the plane of measurements in regard to non-oval shape, but indicates that the definition of round shape and oval shape demands the respective conditions (AP = T or AP < T) to be satisfied on both the transverse and longitudinal planes. One could argue that a nodule that is non-oval on any plane is suspicious according to EU-TIRADS. However, that way there would be two different regimens of measurements for non-oval and round shaped nodules, both types regarded as suspicious in EU-TIRADS.

Irrespectively of these differences, our analyzes, like reports from many other centers, indicate that the assessment of nodule’s suspicious shape is characterized by a very high SPC [10,30,32,33], with lower SEN, especially on the longitudinal plane [30]. SEN of that feature on the longitudinal plane reaches values similar to those on the transverse plane when round nodules are regarded as suspicious too. The inclusion of round nodules into suspicious category improves SEN on the transverse plane by several times, especially in patients with HT. It is noteworthy that the effectiveness of the shape assessment is not improved when the conditions AP > T or AP ≥ T are evaluated on both planes (with positive feature defined as the condition satisfied on any plane) when compared with the measurements performed on a single plane. Both our study and other reports [34] indicate that the assessment on the transverse plane and the longitudinal plane have similar ACC. However, the nodule’s suspicious shape is very rarely identified on the longitudinal plane only. Thus, like Kim et al. (2021) we believe that there is no need for the assessment of a nodule’s shape on both planes and measurements on the transverse plane are satisfactory [32]. There are some earlier contradictory reports in this regard [35].

The highest ACC of the assessment could be obtained with measurements on the transverse plane when the threshold for AP/T ratio is close to 1.15 (1.14 in HT group and 1.17 in non-HT group). If such a threshold is used there are moderate changes in SEN and SPC, but a twofold increase in OR and LR+ is observed when compared to the threshold AP > T. A further increase of AP/T threshold to values proposed by Grani et al. (2020) (AP/T ratio = 1.2) in patients without HT would not improve OR (OR: 4.2 vs. 5.8), and in patients with HT there would even be a loss of discrimination power of the suspicious shape feature (incidence of benign nodules with AP/T ≥ 1.2 did not differ significantly from the incidence of such cancers: 1.1% vs. 3.5%, *p* = 0.1602) [10]. It is possible that these differences between our data and the study by Grani are a consequence of the lack of nodules with indeterminate cytology in the Italian study. In such nodules, US malignancy features generally have lower effectiveness, due to a lower percentage of PTC among cancers. On the other hand, Topaloglu et al. (2016) examined patients with nodules of category III of the Bethesda System for Reporting Thyroid Cytology (BSRTC) and proposed the threshold for AP/T ratio below one (0.81) [11]. A lower threshold was even proposed by Huang et al. (2018) while diagnosing papillary microcarcinomas (0.7) [12]. However, those thresholds were established in a different way. They were optimized to produce maximal SEN and SPC at the same time, but not ACC. When we followed the same priorities then the threshold (as determined with ROC curve analysis) was 0.92 in patients with HT (SEN: 43.7%, SPC: 80.1%, ACC: 71.2%, PPV: 41.3%, NPV: 81.6%, OR: 3.1) and 0.87 in patients without HT: 0.87 (SEN: 46.6%, SPC: 72.5%, ACC: 66.0, PPV: 36.2%, NPV: 80.2%, OR: 2.3). In HT group these values did not differ significantly from those obtained for the threshold AP ≥ T. However, in a non-HT group such a low threshold was unsatisfactory, leading to low SPC and PPV as almost 1/3 of nodules (32.3%) reached that threshold. This is an important weakness in the case of a US malignancy feature that is key in assigning nodules into the highest risk category of SRS. In the studies by Topaloglu and Huang reported SPC values for proposed thresholds were even lower (61.6% and 66.7%, respectively), and PPV did not exceed 30% (Topaloglu et al.: 29.1%, Huang et al.: missing data on PPV) [11,12].

We found that in patients with HT the assessment of nodule’s suspicious shape has low effectiveness in the case of nodules <1 cm. Fukushima et al. (2021) did not find differences in the effectiveness of that feature between small and larger nodules [30]. Ren et al. (2015) found it to be even higher in nodules <1 cm, but they did not analyze the possible influence of HT on that effectiveness [33]. In our group of patients without HT the AUC value for nodules < 1 cm was also slightly higher than for larger nodules.

For obvious reasons, the use of a precise threshold for AP/T ratio in practice demands an additional effort from the ultrasonographer. The authors of some reports admitted that the suspicious shape was determined based on ultrasonographer’s impression instead of the measurements of nodules’ diameters. Such an assessment makes it difficult to detect fine abnormalities in a nodule’s shape. As our data indicate, it is worthwhile to perform proper measurements because even a minor modification of the threshold for nodule’s suspicious shape may lead to a several-fold change in SEN of that feature and marked changes in its SPC and RoM. The local experience of a center is especially valuable in this case, because the reproducibility of the assessment of nodule’s suspicious shape, like other US malignancy features, is not high [36]. It results not only from technical differences in measurements (e.g., caused by different positions of a patient, or different pressure exerted by a probe on the examined area), but also from differences in the epidemiology of thyroid diseases in examined patients (different incidence of HT, other profile of cancers). Thus, in our opinion, it could be advisable to determine the optimal threshold for AP/T ratio in each center individually, to adjust it to its specific situation. It should be noted that while the discussed changes in the AP/T ratio threshold may lead to a several-fold increase in SEN, the resultant SEN is still far from satisfactory, similarly to other high risk features of malignancy.

The results of our study indicate that changes in the threshold of AP/T ratio have a different impact on the effectiveness of particular SRSs in classification of benign and malignant nodules. The impact is bigger for EU-TIRADS than for other SRSs. This is a result of the particular definition of high risk category in EU-TIRADS which is different from other systems based on similar rules: K-TIRADS and ATA guidelines (ACR-TIRADS uses points for US malignancy features unlike all other SRSs). In EU-TIRADS a nodule with strong risk features does not need to be hypoechoic to be classified into category 5 [1]. Consequently, category 5 of that system directly reflects all effects of categorization of all round nodules as suspicious. In addition, these effects are larger in HT group than non-HT group. Similarly, the change of AP > T threshold to a threshold that gives maximal ACC of the nodule’s shape assessment produces more pronounced effects on category 5 in EU-TIRADS than in other SRSs. This effect is advantageous as it leads to larger increase in RoM and SPC of that category. Therefore, in the case of EU-TIRADS there is the greatest possibility of selecting various thresholds of AP/T ratio in order to optimize SPC (the threshold close to 1.15) or SEN (AP ≥ T threshold) of that system.

Despite these differences it should be emphasized that all SRSs showed similar effectiveness in distinguishing benign nodules from cancers in patients with HT at the same thresholds of AP/T ratio. Additionally, there were no significant differences in the effectiveness of those systems in patients with or without HT. Likewise, Wang et al. (2015) did not find any significant influence of HT on the effectiveness of SRSs (they evaluated the system proposed by Kwak, ACR-TIRADS and ATA guidelines) [37]. We showed that EU-TIRADS had the highest SEN with the lowest SPC and ACR-TIRADS had the highest SPC with the lowest SEN irrespectively of the threshold used for AP/T ratio or the presence of HT, which is concordant with other reports [5,37,38,39,40]. In both HT and non-HT patients the calculated RoM of nodules in particular categories of SRSs was generally close to the expected one. Differences over 5% were observed for category 4 of K-TIRADS, ACR-TIRADS and ATA guidelines systems, where the risk in non-HT group was higher than expected. That could be the consequence of a notable fraction of cancers other than PTC in our material, amounting to 23.1% in non-HT group. These cancers were mainly FTC and HTC and were usually classified into category 4 of those SRSs (56.3% cancers other than PTC for K-TIRADS and ATA guidelines, 64.6% for ACR-TIRADS). The higher percentage of cancers other than PTC was also a probable cause of slightly lower AUC of those SRSs in non-HT group than in patients with HT. It should be kept in mind that US malignancy features and SRSs were established mainly on the basis of the ultrasound image of the most common PTCs [41]. Effectiveness of SRSs could potentially be improved by the inclusion of elastographic measurements [42,43]. Promising results in this respect were also obtained in patients with HT [21,44,45].

A limitation of our study is the difference in mean sizes of nodules in patients with and without HT. The size might affect the US characteristics of thyroid nodules, but small nodules are typical of HT. The advantage of our study is performing US malignancy feature evaluation directly prior to biopsy. Therefore, the result of FNA did not influence the evaluation. The majority of final diagnoses were based on postoperative histopathological examination. It is an advantage because of the certainty of the final diagnosis but on the other hand it could be a source of a bias in patient selection. In clinical practice, patients with HT are usually not referred to surgical treatment. Despite this way of including patients in the study the distribution of nodules among equivocal and unequivocal categories of BSRTC was similar in both groups. That is advantageous because there are significant differences in the usefulness of the assessment of US malignancy features in relation to nodule’s category of FNA outcome [6]. The adopted way of confirmation of HT diagnosis may also be regarded as advantageous because of rather rigorous criteria. However, it may be seen as a limitation in relation to seronegative cases of HT. In our study, patients with a positive TPOab test dominated, and seronegative patients, in whom HT was confirmed by its characteristic features in the cytological examination, constituted a very small percentage of HT group.

## 4. Materials and Methods

### 4.1. Examined Patients

Ultrasound imaging and FNA examinations were performed in a single center, in the years 2012–2020, in patients referred by endocrinologists from outpatient clinics. The study included all nodules classified into the categories II–VI of BSRTC with full ultrasound imaging data, a known result of the postoperative histopathological examination and a known status of HT presence. The exclusion criteria consisted of previous surgical or radioiodine treatment, as well as positive neck irradiation history. Because of relatively small number of nodules assigned to category II in BSRTC with a known result of histopathological examination in patients with HT, in that group we additionally included all nodules with full ultrasound imaging data and category II of BSRTC confirmed in at least three FNAs. The study included 1188 nodules (revealed in 1022 patients), i.e., 358 nodules in patients with HT (HT group) and 830 nodules in patients without HT (non-HT group) (Table 5). The differences in the incidence of particular categories of the Bethesda system between groups HT and non-HT did not exceed 5%. Frequencies of nodules with an equivocal (categories III–V) FNA outcome were similar in both groups (HT: 50.8% vs. non-HT: 54.1%, *p* = 0.3018).

In all patients, the clinical diagnosis of HT was established by an endocrinologist in the endocrine outpatient clinic on the basis of clinical symptoms, serological tests (measurement of anti-thyroid antibodies), assessment of serum concentrations of thyroid stimulating hormone and thyroid hormones, ultrasound examination, as well as cytological examinations. Because fewer than 30% of patients with HT present all the above-mentioned HT features [13,14,46], we decided to adopt minimal conditions for the diagnosis of HT. We assumed that all patients in HT group had to have a clinical diagnosis of HT confirmed with elevated levels of serum anti-thyroid peroxidase antibodies (TPOab) or characteristic features of HT in the microscopic examination. On the other hand, patients included in the non-HT group did not have any hormonal, morphological, or ultrasound features of HT and they had normal TPOab.

### 4.2. Analysis of US Malignancy Features

The analysis of US malignancy features was done prospectively. The presence of particular US malignancy features was assessed by experienced sonographers (four physicians with over ten years’ experience) directly before FNA, according to a unified pattern that had been used at our department for many years. We used a computer system dedicated for collecting detailed information on examined nodules in a database. The system was created by one of the authors of the study—MK. On the basis of those data, three diameters of biopsied nodules were determined as well as the presence of: (1) marked hypoechogenicity (compared to the echogenicity of the strap muscles); (2) hypoechogenicity (as compared to the normal thyroid); (3) solid echostructure (>90% solid) (4) more solid than cystic echostructure (>50% solid); (5) suspicious shape/orientation, assessed on the transverse and longitudinal planes, interpreted in two variants: AP ≥ T and AP > T; (6) suspicious margins—irregular (including microlobulated, spiculated, and suggesting extrathyroidal extension); (7) microcalcifications; (8) macrocalcifications; (9) rim calcifications; (10) pathological vascularization (marked intranodular vascular spots). The presence of spongiform echostructure (>50% of nodule, without obvious solid areas) was also assessed. The US examinations were performed with the use of the Aloka Prosound Alpha 7 ultrasound system, ALOKA co. Ltd., Tokyo, Japan with a 7.5–14 MHz linear transducer.

With the use of the set of features specified above, all thyroid nodules were classified into specific categories of four SRSs: EU-TIRADS [1], K-TIRADS [2], ACR-TIRADS [3], and the system recommended by ATA [4]. The differences in the interpretation of nodules with mixed echogenicity between particular SRSs were considered (the presence of any hypoechoic tissue assessed in EU-TIRADS or predominant echogenicity in other SRSs). In the case of the ATA guidelines, a modification was applied, because this system does not cover all nodule’s ultrasound patterns; in particular it lacks patterns in which iso- or hyperechoic nodules show high malignancy risk features. In total, 71 (6.0%) nodules did not satisfy the criteria of ATA classification and those nodules corresponded to 22 cancers (8/9.2% in HT group and 14/6.7% in non-HT group, *p* = 0.4625). We decided to classify such nodules into the intermediate suspicion category. That allowed us to compare how all systems worked in the evaluation of the same set of nodules. We did not identify disrupted rim calcifications with small extrusive soft tissue component as a separate feature (which is included in ATA guidelines), but the nodules presenting such an image were treated as ones with irregular margins which resulted in the same output of the categorization. Two researchers (DSK and MK) independently assigned all the ultrasound features for TIRADS score calculation. In the case of discrepancy, the US report was jointly reevaluated and discussed to confirm its categorization.

### 4.3. Analyses, Statistical Evaluation

At the first step of the evaluation, receiver operating characteristic (ROC) curves were determined in both groups and both thyroid planes for AP/T ratio describing nodule’s shape. The Z test was used to compare the area under the ROC (AUC) value between HT and non-HT groups. The cut-off values of AP/T ratio that showed the highest accuracy (ACC) in the classification of benign and malignant lesions were also identified. Odds ratios (OR) with relative 95% confidence intervals (95% CI) for the established cut-off values were assessed with the use of logistic regression analysis. The effectiveness of the determined thresholds as well as the ‘standard’ thresholds: AP > T and AP ≥ T was described with the use of sensitivity (SEN), specificity (SPC), ACC, positive likelihood ratio (LR+) and the percentage of nodules reaching given threshold. The RoM of those nodules (the proportion of cancers among all positive nodules, i.e., the positive predictive value—PPV) and the negative predictive value (NPV) were calculated. Additionally, the effectiveness of all examined thresholds of AP/T ratio was assessed for nodules <1 cm and larger as well as for cancers other than PTC.

Next, the incidence of other US malignancy features was assessed in the nodules classified into HT and non-HT groups in respect to the division of the nodules into benign lesions and cancers according to the final diagnosis. In the case of nodules of mixed echogenicity the presence of any hypoechoic tissue was regarded as hypoechogenicity. The associations between individual US malignancy features and malignancy were evaluated with the use of logistic regression analysis in both groups. The OR were calculated to determine the relevance of all potential predictors of the outcome. The incidence of all US malignancy features was also compared between benign nodules of HT and non-HT groups and between malignant nodules in both groups.

Then the distribution of benign and malignant nodules among particular categories of the examined SRSs was assessed. The efficiency of the systems was compared analyzing ROC curves and cut-off categories with the highest ACC were identified for each of the SRSs. Using those cut-off categories SEN, SPC ACC, RoM/PPV, NPV were calculated, as well as the percentage of positive nodules in both groups. Those analyzes were performed separately for three variants of AP/T ratio interpretation described above.

The statistical analysis was performed with the Dell Statistica (data analysis software system), version 13, Dell Inc. (2016), Round Rock, TX, USA. The comparison of frequency distributions was performed with chi2 test (with modifications appropriate for the number of analyzed cases). The Kruskal–Wallis test was used for comparing continuous variables between groups. The value of 0.05 was assumed as the level of significance.

### 4.4. Microscopic Examination

Biopsies were performed following regular procedures, on nodules with a diameter of at least 5 mm (and usually over 1 cm) and at least one malignancy risk factor (ultrasonographic or clinical). In most cases, two aspirations of a nodule were done. Smears were fixed with a 95% ethanol solution and stained with haematoxylin and eosin. The FNA outcome of each nodule was classified into one of six categories in the BSRTC—category I: non-diagnostic/unsatisfactory biopsies, category II: benign lesions (BL), category III: follicular lesion of undetermined significance (FLUS)/atypia of undetermined significance (AUS), category IV: suspicious for a follicular neoplasm (SFN), category V: suspicious for malignancy (SM) and category VI: malignant neoplasm (MN) [47,48]. Patients with a FNA outcome of category IV, V, or VI were routinely referred for thyroid surgery. In the case of a diagnosis of BL or FLUS/AUS, surgical treatment was performed based on the patient’s preference or due to the large size of the goiter as well as the presence of other clinical, ultrasonographic or cytological risk features (especially in the case of AUS diagnosis).

## 5. Conclusions

The optimal usage of US malignancy features in patients with HT is very important because of the increased risk of malignancy in thyroid nodules in that group. To reach this goal, it may be helpful to adjust the threshold for AP/T ratio to the specific characteristics of nodules found in patients with HT. This specificity consists in more frequent occurrence of round nodules. In patients with HT, a slight modification of the threshold for AP/T ratio and regarding round nodules as suspicious leads to a three-fold increase in SEN of the suspected nodule’s shape feature, with a disproportionally lower decrease in SPC and similar ACC. Importantly, such a modification is effective also for cancers other than PTC. In patients without HT there are analogical, yet less marked changes. Thus, the use of AP ≥ T threshold instead AP > T is justified, especially in patients with HT and in centers that intend to improve SEN. On the other hand, in centers where, due to the epidemiology of thyroid diseases, the priority is not an improvement of SEN but maximization of SPC and PPV, it is rational to use threshold for the AP/T ratio over 1, close to 1.15. It enables a twofold increase in OR of nodules that reach such a threshold in comparison to the classic AP > T threshold, with a very low loss of SEN and maximization of ACC. It also could lead to an even twofold reduction of the number of performed FNA. The assessment of a nodule’s shape on the transverse plane is sufficient for the effective use of that feature with any of the analyzed thresholds. The diagnostic effectiveness of EU-TIRADS, K-TIRADS, ACR-TIRADS, and ATA guidelines in patients with HT and without HT is similar. Changes in the threshold for AP/T ratio modify that effectiveness, and their influence on the number of nodules classified into the category of the highest risk is the greatest in the case of EU-TIRADS.

## Figures and Tables

**Table 1 cancers-13-04900-t001:** Data on the diagnostic effectiveness of particular thresholds for AP/T ratio in examined groups of nodules (HT and non-HT). Evaluation of AP/T ratio on transverse and longitudinal planes.

The Plane	AP/T Ratio	No./% of Nodules	Ben./Mal *p*	SEN	SPC	ACC	PPV (RoM)	NPV	LR+	OR (95% CI) *p*	AUC (95% CI) *p*
transverse		HT group									
	AP ≥ T	76/21.2	42/34 <0.0000	39.1	84.5	73.5	44.7	81.2	2.5	3.5 (2.0–6.0) <0.0001	0.635 (0.565–0.704) <0.0001
	AP > T	22/6.1	12/10 0.0170	11.5	95.6	74.7	46.6	76.6	2.6	2.8 (1.2–6.7) 0.0212	
	AP/T ≥ 1.14 max ACC	11/3.1	4/7 0.0063	8.0	98.5	76.5	63.6	76.9	5.5	5.8 (1.8–5.4) <0.0001	
		non-HT group									
	AP ≥ T	116/14.0	61/55 <0.000	26.4	90.2	74.2	47.4	78.6	2.7	3.3 (2.2–5.0) <0.0001	0.627 (0.582–0.671) <0.0001
	AP > T	52/6.2	28/24 0.0003	11.5	95.5	74.5	46.2	76.3	2.6	2.8 (1.6–4.9) 0.0005	
	AP/T ≥ 1.17 max ACC	28/3.4	10/18 <0.0000	8.7	98.4	75.9	64.3	76.3	5.4	5.8 (1.8–5.4) <0.0001	
longitudinal		HT group									
	AP ≥ T max ACC	24/6.7	12/12 0.0024	13.8	95.6	75.7	50.0	77.5	3.1	3.5 (1.5–8.0) <0.0038	0.635 (0.572–0.699) <0.0001
	AP > T	9/2.5	7/2 0.8055	2.3	97.4	74.3	22.29	75.6	0.9	0.9 (0.2–4.4) 0.8830	
		non-HT group									
	AP ≥ T max ACC	43/5.2	20/23 <0.0000	11.1	96.8	75.3	53.5	76.5	3.4	3.7 (2.0–7.0) <0.0001	0.647 (0.603–0.690) <0.0001
	AP > T	15/1.8	7/8 0.0108	3.8	98.9	75.1	53.3	75.5	3.1	3.5 (1.3–9.8) <0.0164	

ACC—accuracy, AP—anteroposterior diameter, AUC—area under the receiver operating characteristic curve, Ben.—benign lesion in histopathological outcome, CI—confidence intervals, HT—Hashimoto’s thyroiditis, LR+—positive likelihood ratio, Mal.—thyroid malignancy in histopathological outcome, NPV—negative predictive value, OR—odds ratio, PPV—positive predictive value, RoM—risk of malignancy, SEN—sensitivity, SPC—specificity, T—transverse diameter.

**Table 2 cancers-13-04900-t002:** Comparison of the incidence of sonographic features other than suspicious nodule’s shape in HT and non-HT nodules in relation to the histopathological outcome: benign lesion vs. thyroid malignancy. Results of univariate logistic regression analysis in both groups.

Sonographic Feature	HT Group				Non-HT Group			
	Ben. (271) No/%	Mal. (87) No/%	*p*	OR (95% CI) *p*	Ben. (622) No/%	Mal. (208) No/%	*p*	OR (95% CI) *p*
marked hypoechogenicity *	14/5.2	31/35.6	<0.0001	10.2 (5.1–20.3) <0.0001	33/5.3	60/28.9	<0.0001	7.2 (4.6–11.5) 0.0001
Hypoechogenicity *	148/54.6	75/86.2	<0.0001	5.2 (2.7–10.0) <0.0001	365/58.7	178/85.8	<0.0001	4.2 (2.7–6.3) 0.0001
solid echostructure	247/91.1	84/96.6	0.0965	2.7 (0.8–9.3) 0.1094	436/70.1	188/90.4	<0.0001	4.0 (2.5–6.6) 0.0001
more solid than cystic echostructure	264/97.4	87/100.0	0.2021	-	533/85.7	204/98.1	<0.0001	8.5 (3.1–23.5) <0.0001
suspicious margins	14/5.2	38/43.7	<0.0001	14.2 (7.2–28.2) <0.0001	23/3.7	72/34.6	<0.0001	13.8 (8.3–22.8) <0.0001
microcalcifications	5/1.9	26/29.9	<0.0001	22.7 (8.4–61.4) <0.0001	17/2.7	38/18.3	<0.0001	8.0 (4.4–14.4) 0.0001
macrocalcifications	19/7.0	5/5.8	0.6817	0.8 (0.3–2.2) 0.6822	43/6.9	40/19.2	<0.0001	3.2 (2.0–5.1) <0.0001
rim calcifications	7/2.6	4/4.6	0.5549	1.8 (0.5–6.4) 0.3500	20/3.2	8/3.9	0.6627	1.2 (0.5–2.8) 0.6631
pathological vascularization	37/13.7	13/14.9	0.7628	1.1 (0.6–2.2) 0.7628	119/19.1	48/23.1	0.2192	1.3 (0.9–1.9) 0.2199

*—in the case of nodules with mixed echogenicity the presence of any hypoechoic tissue was considered; Ben.—benign lesion in histopathological outcome, CI—confidence intervals, OR—odds ratio, HT—Hashimoto’s thyroiditis, Mal.—thyroid malignancy in histopathological outcome. Data on nodule’s shape criterion are presented in Table 1.

**Table 3 cancers-13-04900-t003:** Distribution of benign and malignant nodules between particular categories of TIRADS, the comparison of expected RoM with calculated RoM for each category (TIRADS categories corresponding to the lack of nodules have been omitted, nodule’s shape evaluated on the transverse plane).

Category of TIRADS/Guideline	Expected RoM (PPV)	Calculated RoM (PPV)					
		HT Group			Non-HT Group		
		AP ≥ T	AP > T	AP/T ≥ 1.14	AP ≥ T	AP > T	AP/T ≥ 1.17
EU-TIRADS							
2—benign	<3	0.0	0.0	0.0	0.0	0.0	0.0
3—low risk	3–15	4.0	5.1	5.0	7.0	7.5	7.3
4—intermediate risk	15–50	11.8	14.1	15.9	18.2	19.1	19.0
5—high risk	>60	51.5	61.4	62.1	54.6	58.5	62.7
K-TIRADS							
2—benign	0	0.0	0.0	0.0	0.0	0.0	0.0
3—low suspicion	2–4	5.7	6.5	6.2	7,6	8,3	8,1
4—intermediate	6–17	16.0	17.8	19.9	26.1	27.0	27.8
5—high suspicion	26–87	65.5	74.6	74.6	61.6	67.2	69.7
ACR-TIRADS							
1—benign	-	0.0	0.0	0.0	0.0	0.0	0.0
2—not suspicious	<2	0.0	0.0	0.0	3.3	4.8	4.8
3—mildly suspicious	5	5.5	6.7	7.1	9.5	10.0	9.6
4—moderately suspicious	5–20	17.7	20.2	21.2	26.1	27.8	28.7
5—highly suspicious	>20	65.4	76.8	77.4	63.4	69.7	73.9
ATA guidelines							
1—benign	<1	0.0	0.0	0.0	0.0	0.0	0.0
2—very low suspicion	<3	0.0	0.0	0.0	2.5	2.5	2.5
3- low suspicion	5–10	5.6	6.5	6.2	9.4	9.4	9.1
4—intermediate suspicion *	10–20	15.7	18.1	19.8	24.9	26.3	26.9
5—high suspicion	70–90	65.9	74.6	75.0	61.0	66.7	69.9

*—included 71 non-hypoechoic nodules with high risk features (including 22 cancers, 8 in HT group and 14 in non-HT group). AP—anteroposterior diameter, HT—Hashimoto’s thyroiditis, PPV—positive predictive value, RoM—risk of malignancy, T—transverse diameter, TIRADS—Thyroid Imaging Reporting and Data Systems.

**Table 4 cancers-13-04900-t004:** Data on the diagnostic effectiveness of analyzed SRSs in HT and non-HT groups for the high risk category (nodule’s shape evaluated on the transverse plane).

**Index of Effectiveness**	**HT Group**			**Non-HT Group**		
	**AP ≥ T**	**AP > T**	**AP/T ≥ 1.14**	**AP ≥ T**	**AP > T**	**AP/T ≥ 1.17**
	EU-TIRADS					
% of nodules	37.4	28.2	25.7	30.2	26.1	24.2
SEN	79.3	71.3	67.8	65.4	61.1	60.6
SPC	76.0	85.6	87.8	81.5	85.5	87.9
ACC	76.8	82.1	83.0	77.5	79.4	81.1
NPV	92.0	90.3	89.5	87.6	86.8	87.0
AUC (CI 95%)	0.798 (0.747–0.849)	0.817 (0.765–0.869)	0.814 (0.762–0.866)	0.779 (0.744–0.814)	0.782 (0.716–0.818)	0.794 ^a^ (0.759–0.830)
	K-TIRADS					
% of nodules	23.5	18.7	17.6	19.2	15.8	14.7
SEN	63.2	57.5	54.0	47.1	42.3	40.9
SPC	89.3	93.7	94.1	90.2	93.1	94.1
ACC	83.0	84.9	84.4	79.4	80.4	80.7
NPV	88.3	87.3	86.4	83.6	82.8	82.6
AUC (CI 95%)	0.804 (0.749–0.858)	0.808 (0.752–0.764)	0.805 (0.750–0.860)	0.775 (0.740–0.811)	0.775 (0.739–0.811)	0.779 (0.744–0.815)
	ACR-TIRADS					
% of nodules	21.8	15.6	14.8	16.1	11.9	10.7
SEN	58.6	49.4	47.1	40.9	33.2	31.3
SPC	90.0	95.2	95.6	92.1	95.2	96.1
ACC	82.4	84.1	83.8	79.3	79.6	79.9
NPV	87.1	85.4	84.9	82.3	81.0	80.7
AUC (CI 95%)	0.795 (0.741–0.850)	0.791 (0.735–0.848)	0.787 (0.731–0.844)	0.760 (0.724–0.796)	0.752 ^c^ (0.715–0.788)	0.757 ^b^ (0.070–0.793)
	ATA guidelines					
% of nodules	23.7	18.7	17.6	20.0	15.9	14.8
SEN	64.4	57.5	54.0	48.1	42.3	41.3
SPC	89.3	93.7	94.1	89.4	92.9	94.1
ACC	83.2	84.9	84.4	79.0	80.2	80.8
NPV	88.6	87.3	86.4	83.7	82.8	82.7
AUC (CI 95%)	0.809 (0.756–0.863)	0.811 (0.756–0.866)	0.811 (0.757–0.865)	0.768 (0.731–0.804)	0.769 (0.732–0.806)	0.776 (0.740–0.812)

^a^—*p* < 0.01 vs. EU-TIRADS threshold AP > T. ^b^—*p* < 0.05 vs. EU-TIRADS, K-TIRADS and ATA guidelines (threshold AP/T ≥ 1.17 all). ^c^—*p* < 0.05 vs. EU-TIRADS and K-TIRADS (threshold AP > T all). ACC—accuracy, AP—anteroposterior diameter, AUC—area under the receiver operating characteristic curve, Ben.—benign lesion in histopathological outcome, CI—confidence intervals, HT—Hashimoto’s thyroiditis, LR+—positive likelihood ratio, Mal.—thyroid malignancy in histopathological outcome, NPV—negative predictive value, SEN—sensitivity, SPC—specificity, T—transverse diameter, TIRADS—Thyroid Imaging Reporting and Data Systems.

**Table 5 cancers-13-04900-t005:** Data on the diagnostic effectiveness of analyzed SRSs in HT and non-HT groups for the high risk category (nodule’s shape evaluated on the transverse plane).

Parameter	HT Group	Non-HT Group	*p*
Number of nodules	358	830	
Number of patients	310	712	
Age, mean ± SD (years)	55.1 ± 14.0	53.7 ± 13.3	0.1113
No/% of males	12/3.9	94/13.2	<0.0001
Volume of nodules mean ± SD (cm^3^)	3.17 ± 7.4	7.61 ±16.2	<0.0001
No/% of nodules < 1 cm #	58/16.2%	77/9.3%	0.0006
No/% of cancers	87/24.3	208/25.1	0.7813
No/% of PTCs among cancers	76/87.4	160/76.9	0.0411
Other cancers (No/%)	FTC (4/4.6)	FTC (13/6.3), HTC (13/6.3)	
	HTC (1/1.1)	MTC (14/6.7), PDTC (2/1.0)	
	MTC (5/5.7)	AC (2/1.0), ST (2/1.0)	
	ST (1/1.1)	ANG (1/0.5), FT-UMP (1/0.5)	
category of BSRTC (No/%)	II: 124/34.6 *	II: 253/30.5	0.1580
	III: 121/33.8	III: 277/33.4	0.8867
	IV: 40/11.2	IV: 135/16.2	0.0203
	V: 21/5.9	V: 37/4.5	0.2580
	VI: 52/14.5	VI: 128/15.4	0.6925

*—including 78 nodules, without the surgical treatment but after three FNAs with all outcomes classified into category II of BSRTC. #—both cancers and benign nodules <1 cm were more frequent in HT group than non-HT one (34.5% vs. 19.7%, *p* = 0.0068 and 10.3% vs. 5.8%, *p* = 0.0155, respectively). PTC—papillary thyroid carcinoma, MTC—medullary thyroid carcinoma, FTC follicular thyroid carcinoma, HTC—Hurthle cell thyroid carcinoma, PDTC—poorly differentiated thyroid carcinoma, AC—anaplastic carcinoma, ST—secondary tumor, ANG—angiosarcoma, FT-UMP—follicular tumor of uncertain malignant potential.

## Data Availability

The data presented in this study are available on request from the corresponding authors. The data are not publicly available due to patient privacy restrictions.

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
