# Peer review of "Validation of Four Thyroid Ultrasound Risk Stratification Systems in Patients with Hashimoto’s Thyroiditis; Impact of Changes in the Threshold for Nodule’s Shape Criterion"

_cancers, 2021, doi:10.3390/cancers13194900_

Round 1

Reviewer 1 Report

The topic of reliability and value of sonographic features in HT and non-HT patients is of interest and clinical importance.

Generals remarks referring to i. the terminology and ii. concept of the study:

i) The terms "efficacy" and "effectiveness" are employed repeatedly throughout the manuscript. Thus, they might be often used synonymously, they reflect different meanings. I suggest to refer to validity since the manuscript deals with the diagnostic value of US features given by the SRSs.  

ii) Title, Simple Summary and Abstract focus on the "efficacy" of the suspicious shape criterion, only, while major parts of the Results and Discussion section deal with the remaining US malignancy features, as well. The authors anaylse and describe the impact of different AP>T thresholds in both transversal and longitudinal plane in both HT and non-HT and detail the gain / loss of sensitivity dependent from thresholds chosen.  

I suggest to condense the detailed textual presentation of findings in the Results section to avoid duplicate information (Page 3) and to refer to the impact of these threshold changes on global diagnostic value of SRSs in Title, Simple Summary and Abstract, as well. The most accurate shape criterion should be included into Table 2.

Moreover, multiple logistic regression analysis might help to illustrate the proportion of variance attributable to each of the US features.

Detailed comments:

Page 4: The proportion of thyroid  cancer in the study population was high — please comment on .

Page 10, line 328: varying  thresholds of AP/T ratio induce significant changes in sensitivity, which is, however, very low. The limited diagnostic value of a single  US feature should be mentioned.

Legends to table 1 to 4 are missing.

 Someone sentences are confusing and should be rewritten (e.g. Page 8, line 221 to 222, page 10, line 341)

Author Response

Thank you for the appreciation of the clinical importance of our study and for your remarks.

  1. The terms "efficacy" and "effectiveness" are employed repeatedly throughout the manuscript. Thus, they might be often used synonymously, they reflect different meanings. I suggest to refer to validity since the manuscript deals with the diagnostic value of US features given by the SRSs.  

Following that remark, we replaced the term “efficacy” with “effectiveness” as the latter conveys our ideas more precisely. We also employed the term “validation” in the changed title and throughout the manuscript.

  1. Title, Simple Summary and Abstract focus on the "efficacy" of the suspicious shape criterion, only, while major parts of the Results and Discussion section deal with the remaining US malignancy features, as well. The authors anaylse and describe the impact of different AP>T thresholds in both transversal and longitudinal plane in both HT and non-HT and detail the gain / loss of sensitivity dependent from thresholds chosen.  
    I suggest to condense the detailed textual presentation of findings in the Results section to avoid duplicate information (Page 3) and to refer to the impact of these threshold changes on global diagnostic value of SRSs in Title, Simple Summary and Abstract, as well. The most accurate shape criterion should be included into Table 2.

Following this remark, we have modified the title and changed the focus in the Simple summary and in  the Abstract to reflect better the whole set of obtained results. We have also reconsidered the text on Page 3. It has been shortened. Sentences, that described detailed data presented in Table 1, have been deleted.  An additional table has been added to the supplementary material (Table S1) with the comparison of the incidence of round nodules in HT and non-HT groups. Consequently, other supplementary tables have been renumbered.

We do believe that the inclusion of data on the most accurate shape criterion in Table 2 is not an optimal choice. It would lead to the repetition of data presented in Table 1, where thresholds with the highest accuracy are additionally marked as “max ACC” in the “AP/T ratio” column. Instead, we have added the following remainder to the table’s legend: “Data on nodule’s shape criterion are presented in Table 1”

  1. Moreover, multiple logistic regression analysis might help to illustrate the proportion of variance attributable to each of the US features.

The results of multiple logistic regression analysis were presented in supplementary material of the original submission (see Table S4 – previously S3). Independent features in the differentiation between benign and malignant nodules in both groups were enumerated in the section 2.2 of the original version.

Detailed comments:

  1. Page 4: The proportion of thyroid  cancer in the study population was high — please comment on.

The high proportion of cancers in our study is a consequence of rigorous inclusion criteria. More than 93% of analyzed nodules were treated surgically. In the case of HT the indication for the surgical treatment is almost always a suspicious FNA outcome. Thus, we had to additionally include benign nodules with malignancy excluded with triple benign outcome of cytological examination. For these reasons the incidence of cancers in our material does not reflect the incidence of cancers in general population, which is much lower. Additional inclusion criteria (full ultrasound imaging data, a known result of the postoperative histopathological examination and a known status of HT presence) are the reason why the incidence of cancers in our material also differs from the incidence of cancers in operated population. In the final part of Discussion we indicated that the inclusion criteria based on results of postoperative histopathological examination, while offering certainty of the final diagnosis, inevitably “could be a source of a bias in patient selection”.

  1. Page 10, line 328: varying  thresholds of AP/T ratio induce significant changes in sensitivity, which is, however, very low. The limited diagnostic value of a single  US feature should be mentioned.

Following that remark, an appropriate comment has been added at the end of the indicated paragraph.

  1. Legends to table 1 to 4 are missing.

The missing legends to tables in the main manuscript and in supplementary material have been added.

  1. Someone sentences are confusing and should be rewritten (e.g. Page 8, line 221 to 222, page 10, line 341)

Indicated sentences as well as some neighboring sentences have been rewritten.

Reviewer 2 Report

The present study is analysis a very interesting topic - the impact of shape criteria, as one if the five US characteristics used  for risk clasiffication of thyroid nodules – if there is any difference when interpreting this criteria in cases wit hand without HT.

The results are clearly presented , especialy the impact of shape when evaluated in transversal versus anteroposterior section plane.  

Introduction 

  • Please define what do you understand under HT cases – all cases with AITD, or just cases with AITD and increased volume.
  • Also please do clarify the matter – all cases with defined as HT cases had a cytological confirmation of the AITD ( meaning FNAB from the parenchyma)?
  • Please do also refferd to studies comparing the diagnostic power od the most known TIRADS models, even upgraded TIRADS modesl woth elastography:
    • Russ G. Risk stratification of thyroid noduels on ultrasonograăhy with the FRENCH TI RADS – sdecsroption and reflection. Ultrasonography. 2016. 35 (1): 25
    • Borlea A, Borcan F, Sporea I, et al. TI RADS Diagnostic eprformance _ which algorithm is superior ? How elastography and 4 D vascularity improve the malignancy risk assessment. Diagnostics (Basel) 2020. 10 (4):180
  •  

Discussions

If the study did overseen the seronegative  AITD cases (especialy  in atrophic variant of AITD) please do comment about this in the discussions – even stating that the few number of cases will not have a major impact on the results.

Please do comment how come there is no major/significant impact of the final ability of the presented TIRADS models, when there are still significant differences in the AP/T ratio in cases with and withou HT. How would you explaine this phenomenon.

Author Response

Thank you for the positive opinion on our study

Remarks: 

  1. Please define what do you understand under HT cases – all cases with AITD, or just cases with AITD and increased volume.

In the whole study HT cases are considered as all cases that satisfied the inclusion criteria, including the criteria for the diagnosis of HT, with no limitation to AITD cases with increased volume.

  1. Also please do clarify the matter – all cases with defined as HT cases had a cytological confirmation of the AITD ( meaning FNAB from the parenchyma)?

We assumed that all patients in HT group had to have a clinical diagnosis of HT confirmed with elevated levels of serum anti-thyroid peroxidase antibodies (TPOab) or characteristic features of HT in the microscopic examination. On the other hand, patients included into non-HT group didn’t have any hormonal, morphological or ultrasound features of HT and they had normal TPOab.

Characteristic features of HT in the microscopic examination were usually found in the material obtained during FNA of analyzed nodules or other nodules biopsied at the same time, but exclude from the study (e.g. because of missing full US data). So, in general we did not perform FNAB of the parenchyma.

  1. Please do also refferd to studies comparing the diagnostic power of the most known TIRADS models, even upgraded TIRADS models with elastography:
    • Russ G. Risk stratification of thyroid noduels on ultrasonograăhy with the FRENCH TI RADS – decsroption and reflection. Ultrasonography. 2016. 35 (1): 25
    • Borlea A, Borcan F, Sporea I, et al. TI RADS Diagnostic performance which algorithm is superior ? How elastography and 4 D vascularity improve the malignancy risk assessment. Diagnostics (Basel) 2020. 10 (4):180

Some sentences have been added on the prospects of the inclusion of elastography to SRSs and its use in patients with HT (the second last paragraph of the discussion). Appropriate references (including suggested ones) have been added.

  1. If the study did overseen the seronegative  AITD cases (especialy  in atrophic variant of AITD) please do comment about this in the discussions – even stating that the few number of cases will not have a major impact on the results.

We did not analyze seronegative  AITD cases separately. As it was mentioned above “We assumed that all patients in HT group had to have a clinical diagnosis of HT confirmed with elevated levels of serum anti-thyroid peroxidase antibodies (TPOab) or characteristic features of HT in the microscopic examination”. In our material seropositive patients predominated, as there were a few patients with characteristic microscopic features of HT and a negative TPOab test.

We have added some comment on that problem to the final part of the discussion, where limitations of our study are discussed.

  1. Please do comment how come there is no major/significant impact of the final ability of the presented TIRADS models, when there are still significant differences in the AP/T ratio in cases with and without HT. How would you explaine this phenomenon.

The explanation of this phenomenon is complex and we should consider several effects. First, we should note that changes in AP/T ration threshold act in the same direction in patients with or without HT, but to a different degree. Thus, decreasing the threshold and including round nodules leads to the increased sensitivity in both groups, while increasing the threshold to values close to 1.15 (1.14 in HT group and 1.17 in non-HT group) leads to the increased specificity. Second, the suspicious nodule’s shape is only one of several sonographic risk features included in the analyzed TIRADS systems. This very same feature has a different impact on the effectiveness of particular systems. If this effectiveness is measured as the accuracy of classification of benign nodules and cancers into the highest risk category of the system, then, as we showed, the highest impact is observed in the case of EU-TIRADS. That is so, because in systems recommended by ATA and KSThR nodules which are not hypoechoic cannot be classified into the highest risk category regardless of the presence of other high risk features.

Round 2

Reviewer 1 Report

Appropriate changes have significantly

improved the manuscript.